# Inhibition of RNA Binding in SND1 Increases the Levels of miR-1-3p and Sensitizes Cancer Cells to Navitoclax

**DOI:** 10.3390/cancers14133100

**Published:** 2022-06-24

**Authors:** Saara Lehmusvaara, Teemu Haikarainen, Juha Saarikettu, Guillermo Martinez Nieto, Olli Silvennoinen

**Affiliations:** 1Faculty of Medicine and Health Technology, Tampere University, 33520 Tampere, Finland; saara.lehmusvaara@tuni.fi (S.L.); teemu.haikarainen@tuni.fi (T.H.); guillermo.martineznieto@utu.fi (G.M.N.); 2Tays Cancer Center, Tampere University Hospital, 33520 Tampere, Finland; 3HiLIFE, Helsinki Institute of Life Science, University of Helsinki, 00790 Helsinki, Finland; juha.saarikettu@alumni.helsinki.fi; 4Fimlab Laboratories, Pirkanmaa Hospital District, 33520 Tampere, Finland

**Keywords:** inhibitor screen, stress response, RNA binding protein, colon cancer

## Abstract

**Simple Summary:**

Despite of decades of intensive research, several cancer types, for example aggressive colon cancers, are still difficult to treat, and life expectancy is low. Since cancer cells are often resilient and tolerate chemical stresses such as cancer drugs efficiently, they have been difficult to treat. Therefore, combined treatment methods that target cancer cells’ stress tolerance may enhance the treatment outcome. Here we have shown that certain cancer drugs are more effective in colon cancer cells when the expression of a protein called SND1, implicated in regulation of stress responses, is prevented in those cells. We also found that a drug compound called suramin binds to a certain “pocket” of an SND1 protein, and this prevents the interaction of SND1 and certain small RNA molecules, called microRNAs. This block of SND1-microRNA interaction reduces the resilience of colon cancer cells and thus sensitizes them to cancer treatment.

**Abstract:**

SND1 is an RNA-binding protein overexpressed in large variety of cancers. SND1 has been proposed to enhance stress tolerance in cancer cells, but the molecular mechanisms are still poorly understood. We analyzed the expression of 372 miRNAs in the colon carcinoma cell line and show that SND1 silencing increases the expression levels of several tumor suppressor miRNAs. Furthermore, SND1 knockdown showed synergetic effects with cancer drugs through MEK-ERK and Bcl-2 family-related apoptotic pathways. To explore whether the SND1-mediated RNA binding/degradation is responsible for the observed effect, we developed a screening assay to identify small molecules that inhibit the RNA-binding function of SND1. The screen identified P2X purinoreceptor antagonists as the most potent inhibitors. Validation confirmed that the best hit, suramin, inhibits the RNA binding ability of SND1. The binding characteristics and mode of suramin to SND1 were characterized biophysically and by molecular docking that identified positively charged binding cavities in Staphylococcus nuclease domains. Importantly, suramin-mediated inhibition of RNA binding increased the expression of miR-1-3p, and enhanced sensitivity of cancer cells to Bcl-2 inhibitor navitoclax treatment. Taken together, we demonstrate as proof-of-concept a mechanism and an inhibitor compound for SND1 regulation of the survival of cancer cells through tumor suppressor miRNAs.

## 1. Introduction

Cancer is the second most common cause of death in the Western world, and it has been estimated that every third person will get cancer. Despite intensive research and improved therapies, complete eradication of cancer cells is rare in high-grade and metastasized cancers. A common underlying reason is resilience of cancer cells against different stresses during chemotherapy. Cellular stress can be determined as significant disturbances in cell homeostasis, which triggers stress responses aiming to restore homeostasis, or adaptation to changed conditions [1]. Excessive dose of stressor(s) or prolonged exposure may lead to programmed cell death (apoptosis). In cancer cells, rapid proliferation in the non-physiological environment causes different kind of stresses, e.g., hypoxic stress due to the lack of blood supply, oxidative stress and DNA damage stress due to the elevated amounts of reactive oxygen species, and mitotic and proteotoxic stress due to the increased proliferation rate and aneuploidy [2]. In addition, different kinds of cancer treatments cause chemical or physical stress to cancer cells. Therefore, the ability of cells to respond to stress is crucial for cancer cell survival. 

SND1 (Staphylococcal Nuclease and Tudor Domain Containing 1) has been linked to cellular stress response. SND1 is an evolutionary conserved protein consisting of five Staphylococcus nuclease-like (SN) domains and a single Tudor domain (TD). SND1 is overexpressed in several cancers, including colorectal, breast, prostate, and hepatocellular cancers, and gliomas [3,4]. In addition, the expression of SND1 often correlates with the malignant state of the cancer [3,5,6]. SND1 has been shown to employ various mechanisms in normal and malignant cells, and also in stress conditions. SND1, for example, localizes to stress granules, dense protein and RNA aggregations that store untranslated RNA during the stress [7,8,9,10]. In addition, our recent study has revealed that Tudor-SN (SND1)-deficient Drosophila show diminished tolerance to hypoxic stress compared to wild type flies, and the RNA expression profile of SND1 knockout mice livers is practically identical to mice exposed to hypoxia [11]. Moreover, SND1 knockout seedlings of Arabidopsis tolerate salt stress worse than wild type seedlings [12]. In addition, SND1 interacts with several proteins in transcriptional regulation [13,14] as well as in other cellular pathways. For example, SND1-MTDH interaction regulates tumor formation, recurrence, metastasis, and sensitivity to chemotherapeutics in mouse mammary tumor models [15,16]. In addition, SND1-E2F1 interaction reduces the apoptotic activity of E2F1 [17]. Importantly, SND1 binds to RNA via SN domains, and regulates the faith of RNA molecules via different mechanisms. Several studies have demonstrated that SND1 is involved in mRNA splicing [15,18], stability [19] and degradation of adenosine-to-inosine (A-to-I) edited RNA [20], and N6-methyladenosine (m6A) modificated RNA [21,22,23]. SND1 also plays a role in several steps of miRNA-mediated RNAi. First, SND1 functions as a nuclease in RNA-induced silencing complex (RISC) [24], second, SND1 degrades hyper-edited miRNA precursors [20], and third, SND1 mediates the degradation of a certain set of mature miRNAs [25]. SND1 nuclease activity has a template preference, and it degrades miRNAs with CA or UA sites situated between the five most distant nucleotides in mature miRNA sequences [25].

MicroRNAs (miRNAs) are small non-coding RNAs, which regulate many aspects of cell biology, including cell fate determination, stress response, apoptosis and carcinogenesis [26,27]. miRNAs regulate at least 30% of protein coding genes, with an average of 200 mRNA targets per miRNA, and they are frequently dysregulated in cancers. As expected, stress response alters the expression and cellular localization of several miRNAs [1]. Stress response can also modify the expression or activation of adenosine deaminases acting on RNA (ADAR), which further increases the A-to-I edition of miRNA precursors [28]. Editing may affect miRNA degradation and influence miRNA targeting, and thus alter the cellular survival in stress. However, mechanisms of how miRNAs are regulated under stress conditions remain elusive.

In this study, we have demonstrated that in cancer cells SND1 is required to tolerate chemotherapeutic stress, especially drugs targeting apoptosis or the MEK-ERK pathway. By establishing a novel screening assay, we have identified inhibitors for SND1–RNA interaction and shown that the inhibition of RNA-binding sensitizes cancer cells to navitoclax, possibly by preventing the SND1-mediated miRNA degradation of tumor suppressor miR-1-3p.

## 2. Materials and Methods

### 2.1. Cell Culturing

SW480 and HEK-293T cells were obtained from American Type Culture Collection (ATCC) and cultured in Dulbecco’s Modified Eagle´s Medium (DMEM) supplemented with 10% fetal bovine serum, 1% pen-strep, and 1% glutamine in +37 °C, 5% CO_2_. 

### 2.2. ShRNA Constructs, Lentivirus Production and Stable Cell Lines

Silencing of SND1 in the SW480 cells was performed with doxycycline inducible lentiviral shRNA-mediated gene silencing [29]. For cloning the expression vectors for SND1 silencing, the DNA oligo duplex CCGGCGGGATCTCAAGTATACCATTCTCGAGAATGGTATACTTGAGATCCCGTTTTT and AATTAAAAACGGGATCTCAAGTATACCATTCTCGAGAATGGTATACTTGAGATCCCG (forward and reverse strand) for shRNA1, and CCGGTCTCGTCTCAAACTCTATTTGCTCGAGCAAATAGAGTTTGAGACGAGATTTTT and AATTAAAAATCTCGTCTCAAACTCTATTTGCTCGAGCAAATAGAGTTTGAGACGAGA for shRNA2 were inserted into the Age1 and EcoRI sites in the lentiviral transfer plasmid Tet-pLKO-Puro (a gift from Dmitri Wiederschain, Addgene plasmid # 21915). For generation of the control shRNA expression vector, oligos CCGGCCGGAACTCTAGTTTACGATTCTCGAGAATCGTAAACTAGAGTTCCGGTTTTT and AATTAAAAACCGGAACTCTAGTTTACGATTCTCGAGAATCGTAAACTAGAGTTCCGG containing base-substitutions (underlined) compared to the SND1-specific shRNA1 were used. 

Lentiviruses were produced by co-transfecting 293T cells with shRNA containing pLKO-Tet-on plasmid, dR8.91 packaging plasmid and VSV-G envelope protein plasmid in a plasmid ratio of 1:1:0.1, respectively, using FuGENE HD transfection reagent (Promega, Madison, WI, USA). Growth media was exchanged the following day and lentivirus-containing supernatant was harvested 48 h later.

Lentiviral particles were transduced to semiconfluent SW480 cells grown in 6-well plate in the presence of 8 µg/mL polybrene. The day after transduction, the cells were selected for one week in the presence the medium containing 4 µg/mL puromycin (P8833, Sigma-Aldrich, Saint Louis, MO, USA). Optimal shRNA expression and subsequent SND1 silencing was induced by exposing cells to 0.1 µg/mL doxycycline hyclate (D9891, Sigma-Aldrich, Saint Louis, MO, USA) for 7 days. The doxycycline-containing medium was refreshed every second day.

### 2.3. CRISPR-Cas9 Knockout Cell Lines

The SND1 knockout SW480 cell line was established using a double small guide RNA (gRNA) strategy. The target sites were chosen using a CRISPR gRNA design tool (http://crispor.tefor.net/, accessed on 10 October 2018) and two guides were selected, GGTGCGCCATCATTGTCCG and AACGGTTCACATACTATCC, targeting SND1 exon 2 and exon 5, respectively. For SND1 knockout in the HEK-293T cells the GTAGAAAACAAGACTCCCCAG guide RNA was used. Control HEK-293T clones were generated using the GTAGAATACAACACTCGCCAG guide RNA having three base substitutions (underlined) as compared to the SND1-targeting guide RNA. gRNA_Cloning Vector was a gift from George Church (Addgene plasmid # 41824) [30], and pCas9_GFP was a gift from Kiran Musunuru (Addgene plasmid # 44719) [31]. Subsequently, cells were transfected with both plasmids and cultivated in normal growth medium supplied with neomycin (1 mg/mL 5–7 days) (A1720, Sigma-Aldrich, Saint Louis, MO, USA). Growing clones were isolated and genotyped with PCR and Western blot for SND1.

### 2.4. Western Blot and Antibodies

Cells were harvested by scraping into ice cold PBS and lysed using RIPA buffer (150 mM sodium chloride, 1% triton X-100, 1% sodium deoxycholate, 0.1% SDS, 50 mM Tris-HCl, pH 7.5 and 2 mM EDTA) supplemented with protease and phosphatase inhibitor cocktails (vanadate 2 mM, aprotinin 8.3 µg/mL, pepstatin 4.2 µg/mL and PMSF 1 mM). Western Blotting was carried out as previously described using the anti-SN4-1 mouse monoclonal antibody against SND1 [32], or anti-Actin (MAB1501R, Merck Millipore, Burlington, MA, USA). Goat anti-mouse IgG DyLight™ 800 (#35521 Thermo Fisher Scientific, Waltham, MA, USA) was used as the secondary antibody with detection using the LI-COR Odyssey CLx imaging system.

### 2.5. Drug Sensitivity Testing and Drug Responses

Stable SW480 cell lines that conditionally express doxycycline-inducible control shRNA, or SND1-targeting shRNA1 or shRNA2, were treated with 0.1 µg/mL doxycycline for 7 days. Subsequently, cells were plated to 384-well plates in a density of 1000 cells per well. Well plates contained 525 commercially available small molecule inhibitors in five different concentrations. After 72 h, cell viability was measured with CellTiter-Glo (Promega, Madison, WI, USA), and subsequent analyses were carried out as described previously [33]. Drug sensitivity score (DSS), a parameter that analytically integrates the area under the non-linear dose-response model combines the advantages of both the model-based and area-based response calculations. Ref. [34], DeltaDSS (dDSS), which quantifies the response of cancer cells compared to control cells, was calculated by comparing the response of SND1-shRNA cells (shRNA1 or shRNA2) to control shRNA cells.

To validate the screening data, 4000 cells per well were plated to 96-well plates. The following day, the chemotherapeutic drugs were added +/− SND1-RNA binding inhibitors, and 72 h post-drug exposure the cell viability was measured with CellTiter-Glo (Promega, Madison, WI, USA).

### 2.6. Chemotherapeutic Drugs and Cell Viability Assay

For drug sensitivity validation, navitoclax was purchased from MedChemExpress, (South Brunswick Township, NJ, USA), and pimasertib (T6131), trametinib (T2125), and VX-11e (T3166) from TargetMol (Wellesley, MA, USA). Compounds were diluted to DMSO in stock concentration of 10 mM, and stored at −80 °C. Cell viability was measured with CellTiter-Glo (Promega, Madison, WI, USA). 

### 2.7. MiRNA qPCR Panel

Total RNA from SW480 shRNA cell lines treated with doxycycline 0.1 µg/mL for 7 days were extracted with a miRNeasy Mini Kit (Qiagen, Hilden, Germany). To obtain proper biological replicates, the doxycycline treatment was established, and cells were collected for RNA extraction separately for every replicate. A fragment analyzer (Agilent Technologies, Santa Clara, CA, USA) run confirmed the high quality of RNA, which was subsequently reverse transcribed to cDNA with a miRCURY LNA RT Kit (Qiagen, Hilden, Germany). An amount of 20 ng of cDNA was used as a temple for the miRCURY LNA miRNome Human PCR panel (YAHS-301Y, Qiagen, Hilden, Germany), and RT-qPCRs for 372 miRNAs were performed with a BioRad CFX 384 well plate cycler according to the standard protocol of the miRNCURY LNA miRNA PCR Handbook (Qiagen, Hilden, Germany). Threshold for amplification was set to 35 cycles. Data were normalized towards average expression levels of amplified miRNAs, and analyzed with the GeneGlobe analysis program (Qiagen, Hilden, Germany). 

### 2.8. MiRNA qPCR

SW480 cells were treated with 50 µM or 100 µM suramin for 72 h, followed by RNA extraction and reverse transcription as described above (miRNA qPCR panel). Here, three separate cell culture wells represent the replicate samples (n = 3). MiRCURY LNA miRNA PCR primers for miR-103, 340-5p, 582-5p and 1-3p (Qiagen, Hilden, Germany) were used for the RT-qPCR. MiR-103 is commonly used as a reference miRNA as it is normally stably expressed [35]. Additionally, in this analysis the standard deviation of miR-103 was only 0.37 cycles, and thus it was accepted as a reference miRNA.

### 2.9. Protein Expression

SND1 constructs (SND1; 16-888, SN1-SN2; 16-339, SN3-SN4-TSN; 340-888) were cloned by PCR extension cloning to an expression vector with an N-terminal GST-tag followed by a TEV-protease cleavage site. Proteins were expressed in *E. coli* Rosetta 2 (DE3) cells grown in Terrific broth auto-induction media (Formedium, Norfolk, UK) supplemented with 8 g/L glycerol, 30 µg/mL of kanamycin and 30 µg/mL of chloramphenicol. The cells were grown at 37 °C until OD600 reached 1.0 and the temperature was lowered to 22 °C for protein expression. After 16 hours the cells were collected, suspended in lysis buffer (PBS pH 7.4, 0.5 mM TCEP, 1% Triton X-100) and stored at −20 °C.

### 2.10. Protein Purification

Cells were lysed by sonication and the cell debris was cleared by centrifugation (30,000× *g*, 45 min at 4 °C). Supernatant was incubated with Protino^®^ Glutathione Agarose (MACHEREY-NAGEL, Allentown, PA, USA) at 4 °C for one hour and transferred to a gravity-flow column. The column was washed with 50 mL of lysis buffer followed by 20 mL of PBS pH 7.4, 0.5 mM TCEP. TEV-protease was added to the column, and GST-fusion tag was cleaved at 4 °C overnight.

The cleaved proteins were loaded on to a 5 mL HiTrap Heparin HP column (GE Healthcare, Chicago, IL, USA) and the column was washed with 50 mM Bis-Tris (pH 5.5 for SN3-SN4-TSN and pH 6.0 for SND1 and SN1-SN2), 100 mM NaCl, 0.5 mM TCEP. The proteins were gradient eluted against 1.5 M NaCl in the same buffer.

The proteins were further purified with HiLoad 16/600 Superdex 75 (SN1-SN2 and SN3-SN4-TSN) or HiLoad 16/600 Superdex 200 (SND1) pre-equilibrated with 20 mM Tris pH 7.5, 150 mM NaCl, 0.5 mM TCEP. The purified proteins were concentrated and flash-frozen in liquid nitrogen and stored at −80 °C.

### 2.11. Fluorescence Polarization Assay

Fluorescence polarization was measured in 384-well black polystyrene plates (Proxiplate™-384 F Plus, PerkinElmer, Waltham, MA, USA) in a final volume of 10 µL. The reaction was carried out in 20 mM HEPES pH 7.5, 100 mM NaCl and contained 3 nM of fluorescently labeled single-strand RNA (5′ Alexa Fluor^®^ 488 labelled UAGCACCAUUUGAAAUCAGUGUU or 5′ 6-FAM labelled UGGAAUGUAAAGAAGUAUGUAU) and various concentrations of protein. The plate was incubated at 25 °C with shaking at 300 rpm using PST-60 HL plus Thermo Shaker (Biosan, Riga, Latvia) for 1  min. Fluorescence polarization was measured on a PerkinElmer Envision plate reader with FITC FP filters (excitation 480 nm, emission 535 nm). Data were fitted with a one-site binding model using GraphPad Prism version 5.04 for Windows (GraphPad Software, San Diego, CA, USA). 

### 2.12. Screening Assay Development

The screening assay was developed to measure the displacement of the labeled single-strand RNA from SND1. The assay measures the decrease in fluorescence polarization upon RNA dissociation. The reactions were carried out in 384-well plates (Proxiplate™-384 F Plus, PerkinElmer, Waltham, MA, USA) in a total volume of 10 µL. The reactions consisted of SND1, RNA tracer, and test compound in assay buffer (20 mM HEPES pH 7.5, 100 mM NaCl). The plates were incubated at room temperature for 15 min and read with an Envision plate reader (PerkinElmer). DMSO tolerance and the tracer concentration of the assay were optimized. The assay was validated by measuring the repeatability of the maximal and minimal signals between different wells. Forty maximal and minimal signal points were included in the plate and well-to-well variations were calculated as coefficients of variation (CVs). The quality of the assay was measured with common statistical parameters: signal-to-noise ratio (SNR) and screening window coefficient (Z′).

### 2.13. Inhibitor Screening and Potency Measurements

SND1 was screened for inhibition against the LOPAC1280 collection and 1428 nucleotide analogs. Altogether, 2708 compounds were screened at a single concentration (100 µM). The compounds (10 nL) were transferred to the assay plates with an Echo acoustic dispenser (Labcyte, Sunnyvale, CA, USA). Then, 5 µL of SND1 and 5 µL of RNA tracer (final concentrations of 200 nM and 3 nM, respectively) were added to the assay plates. The plates were incubated for 15 min at room temperature before reading with a plate reader. Each screening plate contained blank wells (buffer only), negative controls (0% inhibition) with no inhibitor, and positive controls (100% inhibition) with no SND1.

The potencies of the compounds were measured using half log dilutions of the inhibitors, and the reactions were done in triplicates. Sigmoidal dose–response curves were fitted with four variables using GraphPad Prism version 5.04 for Windows (GraphPad Software, San Diego, CA, USA).

### 2.14. Electrophoretic Mobility Shift Assay

Electrophoretic mobility shift assay (EMSA) was performed by incubating 2 µM of the fluorescently labeled RNA with 2 µM SND1 either with 5 mM EGTA or with 5 mM CaCl_2_ in 20 mM HEPES pH 7.5, 100 mM NaCl at room temperature for 1 or 4 h. EMSA samples were resolved on 20% PAGE gel in 1xTBE buffer with 80 V at 4 °C for 4 h. The fluorescence of the labelled RNA was detected with the Chemidoc MP Imaging System (Bio-Rad, Hercules, CA, USA).

### 2.15. Isothermal Titration Calorimetry

Prior to measurements, SND1 was thoroughly dialyzed against PBS, 0.5 mM TCEP, and suramin was diluted into the dialysis buffer. The binding of suramin to SND1 was analyzed by isothermal titration calorimetry using VP-ITC (MicroCal Inc., Northampton, MA, USA) at 26 °C. The titrations were performed in PBS, 0.5 mM TCEP with 20 μM of protein and 0.25 mM of suramin. Dilution heats were measured and subtracted from the binding isotherms. Data were analyzed with Origin 7 (OriginLab, Northampton, MA, USA).

### 2.16. Differential Scanning Fluorimetry

DSF was performed with protein concentrations of 5 μM. The inhibitor concentrations were 1, 10, and 100 μM. Control wells without the compound were also included. Sypro Orange (Life Technologies, Carlsbad, CA, USA) was used as the reporter dye with a final concentration of 6 × The experiment was performed on a real-time PCR machine (Bio-Rad CFX) with the temperature increasing from 4 °C up to 95 °C with 1° increment per minute.

### 2.17. Molecular Docking

Suramin was docked to two available SND1 crystal structures: SN1-SN2 (pdb code 4qmg) and SN3-SN4-TSN (pdb code 3bdl). Blind docking was performed with Swissdock [36,37] and 31 docked clusters of 8 poses each were generated. Docking results were analyzed with UCSF Chimera [38]. Surface electrostatic potentials were calculated with APBS [39].

### 2.18. Statistical Analysis

A two-tailed *t*-test was used for statistical analysis and the analysis was performed with GrafPad Prism (San Diego, CA, USA) software. 

## 3. Results

### 3.1. Silencing of SND1 Increases the Levels of Tumor Suppressor miRNAs

To investigate the function of SND1 in epithelial cancer cells, we created stable SW480 colon cancer cell lines where doxycycline treatment conditionally and efficiently silenced SND1 (Figure 1A, Appendix A). Silencing of SND1, however, did not reduce cell viability or proliferation rate compared to the control cell line (Figure 1B). SND1 is an RNA binding protein and was recently shown to degrade specific miRNAs through SN domain-mediated interaction [25]. The effect of SND1 silencing on the expression of miRNAs was analyzed in SW480 cells by using a RT-qPCR miRNA panel of 372 miRNAs. The miRNAs were chosen based on previously shown expression in cancer cell lines and their differential expression in disease states (Appendix A). Altogether, 203 miRNAs were amplified in SW480 cell line, and a majority of the differentially expressed miRNAs were upregulated after SND1 silencing (Figure 1C). This result supports the hypothesis that SND1 affects miRNA decay. Interestingly, all of the most upregulated miRNAs (>2-fold), namely miR-1-3p, 133b, 133a-3p, 340-5p, and 582-5p, have been shown to function as tumor suppressors in several cancers [40,41,42,43]. 

### 3.2. SND1 Is Involved in MEK-ERK-Pathway and Bcl-2 Related Apoptotic Pathway

The functional role of SND1 in cancer cells remains elusive, and the SND1-regulated pathways are presently unidentified. Thus, we wanted to analyze the effect of SND1 silencing on the apoptotic effect of 545 cancer drugs that target the main cancer related signaling pathways (Appendix A). In the main drug classes, silencing of SND1 affected most cells that were treated with drugs belonging to apoptotic modulators or kinase inhibitors (Figure 1D–F, Appendix A). Closer analysis of the drug classes revealed particularly high susceptibility in cells treated with Bcl-2 family inhibitors, especially navitoclax or MEK1/2 inhibitors (Figure 1E,F).

Next, we validated the screening results with two MEK-inhibitors (trametinib and pimasertib), one ERK-inhibitor (VX-11e) and one Bcl-2 family inhibitor (navitoclax). The validation data confirmed the synergetic effect of SND1 inhibition (by shRNA) to cell survival with trametinib, pimasertib, VX-11e, and navitoclax (Figure 2A–D). Importantly, similar reduction in survival as with shRNA inhibition was also observed in the CRISPR-Cas9 mediated SND1 knockout cell line (Figure 2E,F, Appendix A).

### 3.3. Discovery of Small Molecule SND1 Inhibitor

To study whether the inhibition of cell survival in response to anticancer drug treatment was mechanistically related to the RNA-binding (and degradation) ability of SND1, we developed a fluorescence polarization (FP)-based inhibitor screening assay for SND1. Currently there are no potent inhibitors for SND1, and the only inhibitor used in cell-based assays, 3′,5′-Deoxythymidine bisphosphate (pdTp), is used at very high (100–200 µM) concentrations [5,25]. We first validated that our recombinant SND1 construct can bind and degrade RNA. We utilized an electrophoretic mobility shift assay (EMSA) to show that SND1 binds, and in the presence of calcium, degrades RNA (Figure 3A, Appendix A). We moved this assay concept to a plate reader setup by utilizing a fluorescence polarization (FP) method, where binding of fluorescently labeled RNA to SND1 increases fluorescence polarization in the absence of CaCl_2_ (Figure 3B). In the presence of Ca^2+^, SND1 degrades the labelled RNA, in line with the Ca^2+^ dependency of its RNase activity, and no increase in polarization can be observed. (Figure 3B). Next, we measured the binding affinity of the SND1 domain constructs towards the RNA tracer (Figure 3C). 

From the results it became evident that full-length SND1 is required for high affinity interaction with RNA, and the two shorter constructs (SN1-SN2; 16-339, SN3-SN4-TSN; 340-888) have ~100-fold lower binding affinities. Thus, we used the full-length SND1 construct to develop an assay that measures RNA-binding to SND1 (Figure 4A). The assay parameters were optimized and DMSO tolerance was determined to be sufficiently high for small molecule screening (Appendix A). This assay was used to screen the Sigma LOPAC^®^1280 library consisting of 1280 pharmacologically active compounds (Figure 4B). Since bacterial staphylococcal nucleases are inhibited by a nucleotide analog pdTp [44], which has also been used as a SND1 inhibitor, we also screened a hand-picked nucleotide analog library. Several hits were identified from the screens, and the most potent ones were found to be known P2X purinoreceptor antagonists (Figure 4C). P2X receptors are cation-permeable membrane ion channels, which open in response to binding of ATP. Three hit compounds with the highest potencies from initial analysis (suramin, NF 023, and PPNDS) were further characterized with the FP assay against the three SND1 constructs (Figure 4D–F, Table 1). The compounds inhibited RNA-binding to full-length SND1 and to both SND1 fragments, indicating multiple binding sites within the full-length protein. The binding of these compounds to SND1 was further validated with differential scanning fluorimetry (DSF) (Figure 4G). Full-length SND1 showed stabilization with NF 023 and destabilization with PPNDS. The SN1-SN2 construct was destabilized by suramin, while other compounds showed no effects on their thermal stability. The SN3-SN4-TSN construct was highly stabilized by suramin and NF 023, while PPNDS displayed a small destabilizing effect. Taken together, all three compounds affected the thermal stability of the SND1 constructs, suggesting compound-induced conformational effects, and interestingly marked differences in stabilization and destabilization were evident between the two SND1 fragments. Full-length SND1 displayed only minor shifts, which could be due to multiple simultaneous thermal melts partially masking the effects arising from single domains. 

As suramin displayed the highest potency in the FP assay and the largest thermal stabilization/destabilization effects, we further analyzed the binding of suramin to the SND1 constructs with isothermal titration calorimetry (ITC) (Figure 4H,I, Table 2). However, we could not obtain a good binding isotherm with full-length SND1–suramin titration due to the high stoichiometry of the interaction. The SN1-SN2 and SN3-SN4-TSN fragments bound suramin with modest affinity, comparable to IC50 values obtained from FP. The data were fitted both with one-site and two-site binding models, since the number of binding sites was unknown. Strong enthalpy changes were associated with suramin binding to both fragments, and the binding events were enthalpically driven. Due to the negative effect of the entropy term, binding affinities remained in the micromolar range.

### 3.4. Characteristics of Suramin Binding to SND1

To identify the potential binding sites for suramin, we docked suramin to available crystal structures of SN1-SN2 (pdb code 4qmg) and SN3-SN4-TSN (pdb code 3bdl) (Appendix A). With SN3-SN4-TSN, a majority of the docked poses (29 of the 31) clustered into two cavities on the protein surface (Appendix A). Poses clustered in a binding cavity in the SN3 domain (23 clusters) were located at a positively charged region overlapping the predicted binding site for RNA [45]. The other docked poses (6 clusters) resided in the TSN domain, in a mainly positively charged region. Docking of suramin to SN1-SN2 domains did not lead to as highly clustered poses as with the SN3-SN4-TSN fragment. One side of the SN1-SN2 fragment was highly positively charged and was where most of the poses were located. Two main clusters of poses were identified, one containing 7 poses (Appendix A) located on a positively charged groove between SN1 and SN2 domains. The other cluster containing 11 poses was located on a positively charged groove perpendicular to the SN1-SN2 interface and running along the surface of both SN domains. Thus, the binding cavities of these two clusters are partly overlapping. Importantly, the MTDH binding site in the SN1-SN2 fragment is located on the opposite side of the suramin binding sites [46]. The large positively charged surface on SN1-SN2 fragment explains the high stoichiometry of suramin interaction observed with ITC: suramin carries highly negatively charged naphthalene trisulfonic acid groups capable of forming interactions along the positively charged surface leading to several potential differential binding sites.

### 3.5. Suramin Treatment Increases the Expression of miR-1-3p

Next, we validated the most prominent hits for SND1 (Table 1) in cells and discovered that suramin decreases the survival of navitoclax-treated SW480 cells in a similar manner as the silencing of SND1 (Figure 5A). Other hits, NF 023 and PPDNS, however, caused no synergetic reduction to cell viability (Figure 5A, Appendix A). Interestingly, the effect of suramin seems to be navitoclax-specific since the survival effect was not detected with MEK-inhibitor pimasertib, although shRNA-mediated silencing of SND1 did show a synergistic effect (Figure 5B). We ruled out the possibility that navitoclax has a direct effect on SND1-RNA-interaction using the FP assay (Appendix A). For cell survival, suramin treatment alone has only a slight effect, but in combination with navitoclax, the cancer cell viability is reduced from 100% to less than 15% (Figure 5C). Suramin treatment also enhanced the amount of tumor suppressor miR-1-3p and the miRNA was shown to bind SND1 with high affinity also in vitro (Figure 5D, Appendix A). We also verified that suramin inhibits miR-1-3p binding to SND1 specifically (Appendix A). Finally, the dependency of miR-1-3p expression on SND1 and the sensitivity to suramin was confirmed in an independent cell line. As SND1 miRNA nucleolytic activity has previously been demonstrated in HEK-293T cells, the same cell line was chosen for this experiment [25]. The expression of miR-1-3p was markedly elevated upon SND1 deletion in HEK-293T cells and suramin resulted in increased expression of the miRNA (Figure 5E,F).

## 4. Discussion

In this study, we propose a novel mechanism for how SND1 affects the survival of cancer cells. We demonstrated that inhibition of the RNA-binding ability of SND1 leads to increased sensitivity of cancer cells to Bcl-2 family inhibitor navitoclax. This increased sensitivity to apoptosis is linked to the expression of tumor suppressor miRNA miR-1-3p.

In cell viability assay, MEK1/2 inhibitors produced the most significant reduction in viability in SND1-deficient cell lines compared to control cells. Interestingly, also all ERK1/2 inhibitors impaired cell viability in these cells, although the effect was not as high as with MEK1/2 inhibitors (Figure 1E). BRAF inhibitors, however, did not decrease viability of SND1 deficient cells compared to control cell line, probably because BRAF is not mutated in the SW480 cell line [47]. Instead, in the SW480 cell line, Ras has an activating G12V mutation, which can have an effect on the expression levels and activity of MEK and ERK. In addition to MEK-ERK pathway inhibitors, also all mTOR inhibitors reduced the survival of SND1-deficient cell lines. PI3K inhibitors, however, did not induce differential survival in SND1-deficient cell lines, and PI3K/mTOR inhibitors only partly. These data suggest that SND1 may play a role in mTOR-mediated signaling pathways in PI3K independent manner. Further studies are needed to confirm the precise mechanism. 

Interestingly, silencing of SND1 enhances the efficiency of Bcl-2/Bcl-XL dual-inhibitor navitoclax, but not Bcl-2 specific venetoclax, or Bcl-XL specific WEHI-539. This may indicate the need for dual targeting of both Bcl-2 and Bcl-XL to obtain the efficient decline in cancer cell viability. Since Bcl-XL-mediated thrombocytopenia is an unfavorable side-effect of navitoclax, smaller navitoclax doses with higher efficiency would be beneficial [39]. Combined treatment of navitoclax and SND1 inhibitors could reduce the required dose of navitoclax for clinical response.

Suramin is a polypharmacological agent used to treat the sleeping sickness caused by Trypanosoma brucei. Suramin also has a range of antiviral and anticancer properties [48]. In this study, we found that suramin inhibits the RNA-binding activity of SND1. Suramin treatment did not impair cell viability with MEK-ERK pathway inhibitor pimasertib in a similar manner as with navitoclax while silencing of SND1 synergized with pimasertib. The lack of synergetic effect of suramin with pimasertib suggests that, in the MEK-ERK pathway, RNA binding might not be a key mechanism of action for SND1. Indeed, SND1 has been shown to employ several other functions including protein-protein interactions, e.g., with MTDH, transcriptional effects, or Tudor domain related demethylated ribonucleotide binding [14,17,21]. Thus, the multiple functions of SND1 may provide possibilities to affect distinct cellular functions by targeting a specific activity of SND1.

miR-1-3p is one of the most studied miRNAs and it has been shown to have critical functions in normal development and physiology of muscle tissues [49]. In addition, the expression of miR-1-3p is reduced in various types of cancer, and dozens of studies point to the tumor suppressor role of the miR-1 (reviewed by [50]). miR-1 targets several cancer-related molecules, such as Slug, PI3KCA, FoxP1, Pim-1, HDAC4, CCDN2, and CXCR4, indicating that miR-1-3p regulates the expression of several oncogenic pathways. Interestingly, upregulation of miR-1 has been shown to decrease the levels of Bcl-2 in several cancers, including breast, prostate, lung, and colorectal cancers [51,52,53,54], and recently, Chen et al. have discovered the direct miR-1 binding site from the Bcl-2 3′UTR region [52]. In addition, miR-1-3p has been associated to chemo- or radiosensitivity in gastric, lung, breast, and colorectal cancers [51,52,53,55]. In the present study, we discovered that miR-1-3p overexpression sensitizes cells to navitoclax treatment. miR-1-3p has indeed interesting therapeutic potential, but delivering miRNAs to cells upon systemic administration faces serious challenges. Thus, increasing the abundance of miR-1-3p by inhibiting SND1-miRNA interaction may have potential in novel therapies.

Suramin was identified in a novel screening assay as an inhibitor of SND1 RNA-binding and the compound was biophysically characterized using DSF and ITC. Suramin and other compounds identified in this study differ mechanistically from previously identified SND1 inhibitors, such as the generic staphylococcal nuclease inhibitor, pdTp [3], and recently identified compounds inhibiting SND1-MTDH interaction [16]. Potential suramin binding sites were identified through blind docking to SND1 crystal structures. These sites where located on positively charged cavities that are well-suited for interaction between negatively charged suramin and native SND1 ligand, RNA. However, due to its polypharmacological activity and modest IC50, it does not represent an ideal candidate for further optimization. It is also likely that other hits found in the screening, NF 023 and PPNDS, bind to these sites. These inhibitor binding sites did not overlap with the SND1-MTDH binding site [46]. These results and the results with pimasertib highlight the differences obtained in cell viability assays when SND1 is silenced vs. when RNA binding is inhibited: silencing leads to total abrogation of SND1 function whereas selective inhibition only affects the RNA-binding and hydrolysis functions of SND1 (Appendix A).

Despite the recent advances in molecular medicine, intrinsic or obtained chemoresistance remain a major challenge for cancer treatment. Due to the high proliferation rate, cancer cells experience additional and altered stresses compared to normal cells. Therefore, combined targeting of stress pathways together with apoptosis pathways can lead to more efficient cancer treatments with reduced side effects and less drug resistance. Inhibition of stress support pathways can sensitize cancer cells to apoptosis, that consequently can reduce the toxicity of the treatment. Indeed, the normal life span and general well-being of SND1−/− mice indicates that an SND1 inhibitor could be well-tolerated and targeting SND1 could cause only minimal side-effects [11]. 

This study contains limitations that require further studies. For example, the target of miR-1-3p is not studied here. However, a recent study has revealed bcl-2 as a direct target for miR-1 [52], although several other cancer-related pathways are also targeted [50]. It is also important to keep in mind that since SND1 has multiple functions, the altered miRNA expression levels and cell viability effects detected after SND1 silencing may be caused by other functions than miRNA binding and degradation. While some of these functions are subject to inhibitor development [16], the importance of other functions for cancer remains to be determined. The present studies were conducted with colon carcinoma cells and it will be important to extend similar studies to other cancer types.

Taken together, we have here demonstrated that silencing of SND1 increases the level of several tumor suppressor miRNAs and sensitizes cancer cells especially to MEK-ERK pathway inhibitors and Bcl-2/Bcl-XL inhibitor navitoclax. Inhibiting SND1-RNA interaction with suramin increases the abundance of miR-1-3p and navitoclax-mediated chemosensitivity. Although suramin is not an optimal inhibitor for SND1, the established screening assays provide tangible tools for more extensive approaches for identification of effective SND1 inhibitors. Taken together, SND1 is an interesting novel drug target, where inhibition of RNA-binding can cause specific, miRNA-mediated chemosensitization. 

## 5. Conclusions

Our study shows that SND-1 regulates the abundance of several tumor suppressor miRNAs, including miR-1-3p, that may affect tumor progression. Indeed, inhibiting SND1’s RNA binding capability with novel SND1 inhibitor suramin sensitizes cancer cells to navitoclax treatment. Thus, specific inhibition of SND1 RNA binding may enhance the effects of conventional chemotherapeutics.

## Figures and Tables

**Figure 1 cancers-14-03100-f001:**
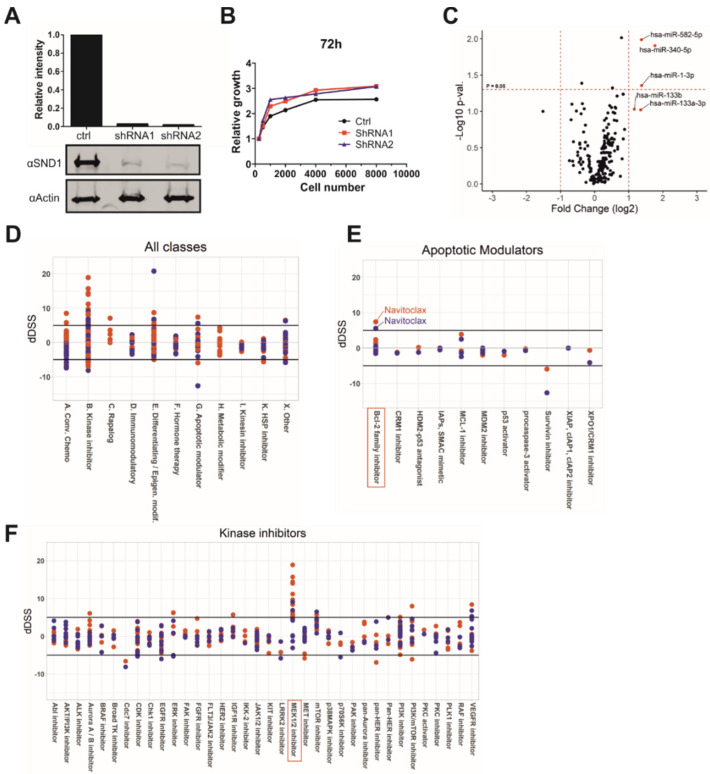
Silencing of SND1 induces the expression of tumor suppressor miRNAs and sensitizes cancer cells to certain anticancer drugs. (**A**) Western blot illustrating the silencing efficiency of stable cell lines, which express doxycycline-inducible shRNA constructs targeting SND1 (shRNA1 and shRNA2) or non-targeting control shRNA (ctrl). (**B**) Growth curve of stable cell lines after doxycycline treatment. CellTiter-Glo assay detected the amount of cell metabolism after 72 h from seeding of different cell densities (250–8000 cells). (**C**) Expression changes of 203 miRNAs, which expression was detected with RT-qPCR from SND1 silenced (shRNA1 and shRNA2) and control SW480 cell lines after doxycycline treatment. For statistical analysis, miRNA expression results were combined from shRNA1 and shRNA2 and the expression changes were compared to control shRNA cell line. N for each group (shRNA1, shRNA2 and control) is three. (**D**–**F**) The screen of 545 cancer drug compounds reveals the chemotherapeutic drug classes (**D**) and specific pathways from main drug classes (**E**,**F**), which sensitize SW480 cells to SND1 silencing. DeltaDSS (dDSS) presents the viability changes compared to control shRNA expressing cell line. dDSS value above five can be considered as a significant change. Red dots: shRNA1, dark blue dots: shRNA2.

**Figure 2 cancers-14-03100-f002:**
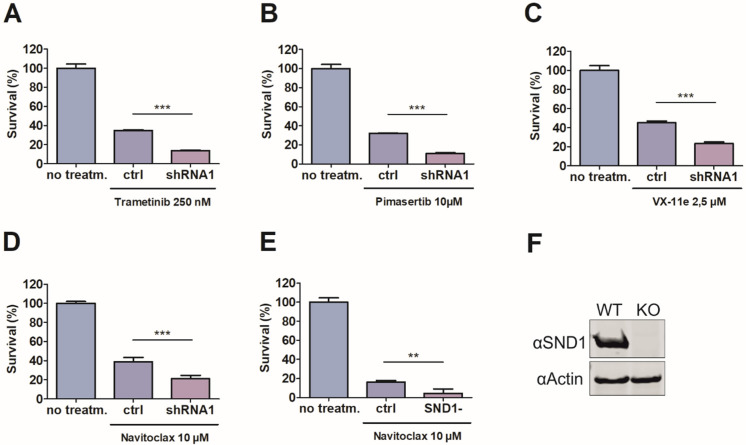
SND1 deficiency sensitizes cancer cells to MEK-ERK pathway inhibitors and Bcl-2 family inhibitor navitoclax. (**A**–**D**) CellTiter-Glo assay measuring the viability of the cells after indicated drug treatment (72 h) in control and stable, conditionally SND1 silenced cell lines (shRNA1). (**E**) Viability reduction after navitoclax treatment (72 h) in control and CRISPR-Cas9 mediated SND1 knock out cell line. (**F**) Western blot illustrating the absence of SND1 protein in SND1 knock out cell line. Two-tailed t-test was used for statistical analysis. Error bars indicate the standard error of the mean. N = 3, ** = *p* < 0.01, *** = *p* < 0.001.

**Figure 3 cancers-14-03100-f003:**
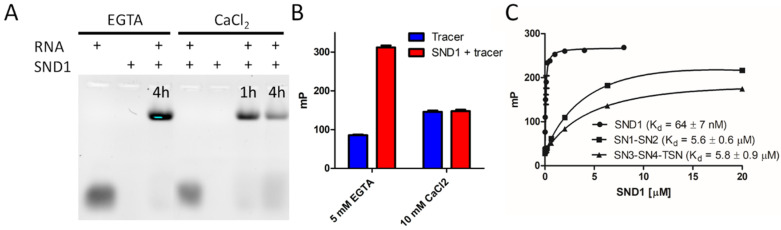
SND1 binds and cleaves fluorescently labelled RNA. (**A**) RNA binding and degradation of SND1 analyzed by EMSA. An amount of 2 µM of fluorescently labeled RNA was mixed with 2 µM SND1 either with 5 mM EGTA or with 5 mM CaCl_2_ and incubated for 1 or 4 h followed by separation with 20% PAGE. The fluorescence of the miRNA was detected with an imager. Controls consisting of 2 µM RNA or 2 µM SND1 alone were separated on the gel without prior incubation. (**B**) FP assay using the same fluorescently labelled RNA tracer. Binding of the tracer to SND1 increases polarization in the presence of EGTA but not in the presence of CaCl_2_. (**C**) Binding affinities of SND1, SN1-SN2, and SN3-SN4-TSN to the RNA tracer measured with the FP assay.

**Figure 4 cancers-14-03100-f004:**
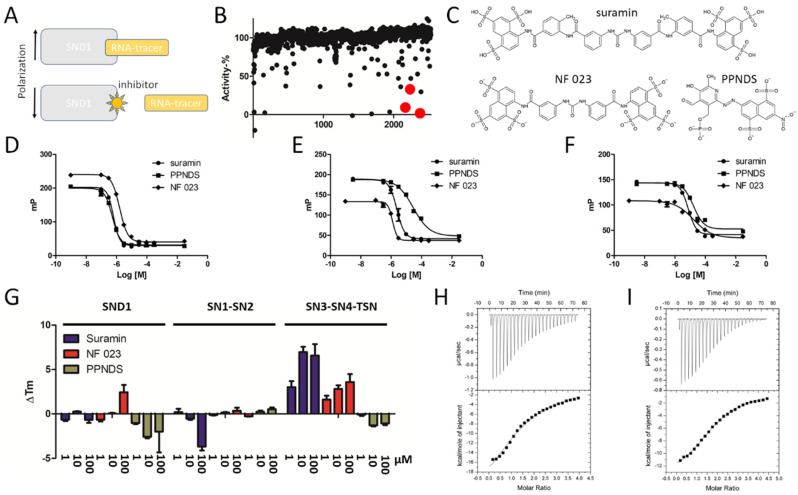
Inhibitor screening and validation. (**A**) Principle of the FP-based screening assay. Binding of an inhibitor displaces RNA-tracer and decreases polarized fluorescence emission. (**B**) Inhibitor screening. LOPAC 1280 library and handpicked nucleotide analogs were screened with the FP assay. Suramin, NF 023, and PPNDS are marked as red spheres. (**C**) Chemical structures of three hit molecules. (**D**–**F**) Potency measurements of the hits against SND1 (**D**), SN1-SN2 (**E**), and SN3-SN4-TSN (**F**). (**G**) DSF validation of the inhibitor binding. (**H**,**I**) ITC of suramin against SN1-SN2 (**H**) and SN3-SN4-TSN (**I**).

**Figure 5 cancers-14-03100-f005:**
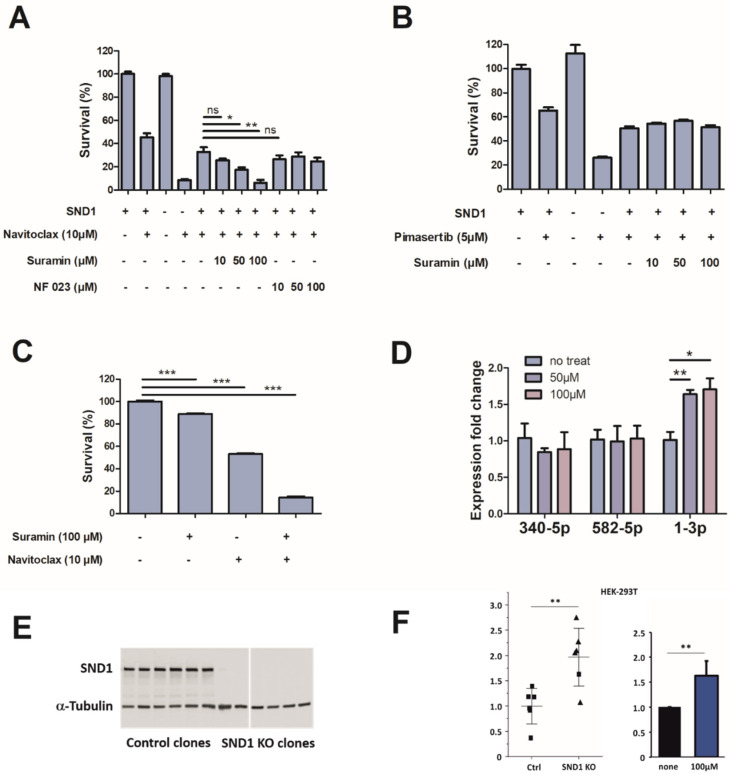
Suramin sensitizes cancer cells to navitoclax treatment and increases the level of miR-1-3p. (**A**–**C**) CellTiter-Glo assay representing the viability differences in conditionally SND1-silenced SW480 cell line (shRNA1), or after indicated drug trea tment (72 h). (**D**) RT-qPCR of most significantly over expressed miRNAs after SND1 silencing. Suramin treatment (72 h, 50 µM or 100 µM) increases miR-1-3p expression levels compared to non-treated control. Two-tailed *t*-test was used for statistical analysis. Error bars indicate the standard error of the mean. N = 3. (**E**) Western blot from HEK-293T SND1 knockout (KO) and control cell clones illustrate absence of SND1 protein in KO clones. (**F**) Left-side graph: Expression of miR-1-3p in HEK-293T SND1 KO and control cells. The expression was quantified by RT-qPCR from six control and KO cell lines by normalization to miR-103-3p expression. Each data point represents the mean normalized expression from four independent experiments. The horizontal lines show the mean expression of the KO and control and the error bars indicate the standard deviation. Right-side graph: The effect of 100 µM suramin on miR-1-3p expression in the parental HEK-293T cells. Two-tailed t-test was used for statistical analysis. Error bars indicate the standard deviation. N = 3. Ns = nonsignificant, * = *p* < 0.05, ** = *p* < 0.01, *** = *p* < 0.001.

**Table 1 cancers-14-03100-t001:** Potencies of the hit compounds.

Protein Construct	IC_50_ ± Std. Error (µM)		
	Suramin	NF 023	PPNDS
SND1	0.6 ± 1.0	1.6 ± 1.0	0.5 ± 1.0
SN1-SN2	2.4 ± 1.1	1.2 ± 1.1	28.4 ± 1.1
SN3-SN4-TSN	7.5 ± 1.1	16.3 ± 1.3	16.5 ± 1.1

**Table 2 cancers-14-03100-t002:** Thermodynamic parameters of suramin binding to SN1-SN2 and SN3-SN4-TSN with ITC.

Thermodynamic Parameters	SN1-SN2	SN3-SN4-TSN
One-site binding model		
K (M^−1^)	8.24 ± 1.09 × 10^4^	1.79 ± 0.09 × 10^5^
ΔH (kcal/mol)	−28.8 ± 1.1	−14.8 ± 0.3
−TΔS (kcal/mol)	22.0	7.7
ΔG (kcal/mol)	−6.8	−7.1
n	1.5	1.9
Two-site binding model		
K_1_ (M^−1^)	2.45 ± 0.33 × 10^6^	7.04 ± 6.97 × 10^5^
ΔH_1_ (kcal/mol)	−16.2 ± 0.1	−11.9 ± 0.1
−TΔS_1_ (kcal/mol)	7.5	3.9
ΔG_1_ (kcal/mol)	−8.7	−8.0
n_1_	0.9	1.0
K_2_ (M^−1^)	5.96 ± 1.01 × 10^4^	8.47 ± 4.08 × 10^4^
ΔH_2_ (kcal/mol)	−16.6 ± 0.3	−18.0 ± 2.1
−TΔS_2_ (kcal/mol)	10.1	11.2
ΔG_2_ (kcal/mol)	−6.5	−6.8
n_2_	1.8	1.0

## Data Availability

The data is maintained in this article.

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
