# Peer review of "Inhibition of RNA Binding in SND1 Increases the Levels of miR-1-3p and Sensitizes Cancer Cells to Navitoclax"

_cancers, 2022, doi:10.3390/cancers14133100_

Round 1
Reviewer 1 Report
This paper describes the identification and potential use of a pharmacological inhibitor of SND1 as a way of sensitizing cancer cells to the chemotherapy drug Navitoclax. Utilizing SW480 cells in which SND1 has been knocked down, they show that SND1 reduction can sensitize SW480 cells to several different chemotherapies. Knockdown of SND1 also upregulates many miRNAs, presumably due to the fact that the protein contains 5 Staphylococcus nuclease-like (SN) domains that can mediate miRNA decay. Given that the five most highly upregulated miRNAs have been previously associated with tumor suppression, the authors performed a fluorescence polarization screen to identify inhibitors that might interfere with SND1 RNA binding. From this screen, they identify several inhibitors, 3 of which they examine further. Of these, suramin, a known P2X purinereceptor antagonist, showed the most promise and was further investigated. They show that suramin, in high doses, appears to sensitize SW480 cells to Navitoclax, although not to Pimasertib, another drug that was observed to be sensitized by SND1 knockdown. Suramin treatment, at high doses, upregulates one of the miRNAs seen to be upregulated by SND1 knockdown in SW480 cells, and this upregulation appears to be repeated in the human embryonic kidney line, HEK-293 cells.
The work demonstrating that SND1 reduction enhances sensitivity to certain chemotherapy drugs is convincing, and, coupled with the upregulation of tumor suppressor miRNAs, seems a reasonable justification for the inhibitor screen. Unfortunately, the only inhibitor that came out of the screen and shows any synergy with a drug is suramin, which is only effective at very high doses (50-100 uM). The authors themselves appear to dismiss another, previously identified inhibitor, pdTp, due to its effectiveness only at very high concentrations. The fact that suramin is highly negatively charged appears to be the driving force in its association with SND1, that has multiple positive patches. This association, however, as suggested by the multiple possible models for binding that the authors present, appears to be of very low specificity, which may be at least part of the reason that such high concentrations are required to see effects. This significantly limits the enthusiasm that this inhibitor could somehow be optimized by at least two orders of magnitude to bring it into the realm in which it could be considered therapeutic (and not likely to have off-target effects).
The connection between the interaction of suramin with SND1 and the effect on RNA binding is also not well established in the paper:
- The authors show that recombinant SND1 shifts a certain short RNA in an EMSA assay, but they do not show 1) that other RNAs are not shifted, and 2) that suramin inhibits this binding/mobility shift. These experiments are important in terms of understanding the specificity of the RNA binding and the suramin inhibition.
- In addition, the authors show in the first part of the paper that SND1 knockdown sensitizes the cells to Pimasertib treatment, but suramin does not recapitulate this effect. If, as the authors seem to be suggesting, the suramin effect is somehow linked to upregulation of miRNA 1-3p (the only miRNA that appears to be upregulated by suramin, even though others are upregulated by SND1 KD), then it would be nice to see what happens to miRNA 1-3p in these Pimasertib + suramin cells.
- The authors show that SND1 KD upregulates 5 miRNAs over two-fold, and yet at least two of these miRNAs are not upregulated by suramin treatment. Do these data correlate with gel shift data, i.e., are these miRNAs bound by SND1, and does suramin disrupt binding specifically to 1-3p, that is upregulated upon treatment of cells, but not binding to the others, that are not?
- In suramin-incubated, Navitoclax-treated cells, would a molecular sponge that inhibits miRNA 1-3p make the cells resistant again? This would go a long way to arguing that the effects are specific.
Minor questions/comments:
- In figure 3B, I assume that what is being compared in the EGTA samples and CaCl2 samples is tracer alone vs. tracer plus SND1 – this is not clear from the figure. Also, in this panel, it is not clear what the significance is of the FP in the presence of CaCl2, where the RNA is presumably being degraded. If there is not the same amount of fluorescent probe in the samples being compared, it seems to me that the polarization shift is not meaningful.
- In figure 5, the statistical significance of the different treatments should be indicated for all of the experiments. Particularly in the suramin – Navitoclax experiments, it is not clear that the slightly lower survival rates in the two lower concentrations of suramin are statistically different from the Navitoclax alone sample.
Author Response
Thank you for your contribution to review-process. Please find our response attached.

Reviewer 2 Report
The article "Pharmacological inhibition of RNA binding in SND1 increases the level of miR-1-3p and sensitizes cancer cells to navitoclax” describes a mechanism and an inhibitor compound for SND1 regulation of the survival of cancer cells through tumor suppressor miRNAs. In general, the article is interesting but I have some major concerns:
- In the present study, the authors have demonstrated that navitoclax functions as an inhibitor in the SND1 regulatory mechanism. In Figure 2, the authors observed the survival of cancer cells according to the concentration of novitoclax in both SND1 silenced cell lines (D) and knockout cell lines (E). What is the reason for the different concentrations of Navitoclax treated in the two different cell lines despite almost no expression of SND1 as a result of Western blot? Did you see results with the same concentration of Navitoclax in both cell lines?
- The authors demonstrated that suramin treatment increased the expression of the tumor suppressor miR-1-3p. However, after the treatment of suramin, the expression level of miR-1-3p did not change significantly to less than two times (Figure 5D). Do the authors think that suramin-modulated miR-1-3p expression level affected the function of navitoclax? There is no functional analysis of miR-1-3p. It should be necessary that some experiments to prove the direct relationship between the regulatory mechanism of SND1 on cancer cell survival and tumor suppressor miRNAs can be performed.
Author Response

(The authors gave the same response as above.)

Reviewer 3 Report
The current manuscript is about drug sensitivity via focusing signaling network interactions. The quality of all figures should be improved and their symmetry should be considered. They are not attractive at current form. A schematic figure of molecular pathways discussed in the current manuscript should be added. Add limitations of current work in conclusion. It is surprising that title of current work is new, but there is only one reference from 2020. Look at literature for relevant works and cite and discuss them in your paper. Such action is vital for promoting visibility and quality of your work and showing that it is a new work. The introduction should be started with cancer malignancy, epidemiology and ability of tumor cells in drug resistance development. I could not see any general discussion of miRNAs in introduction or discussion sections. As miR has been mentioned in title, it is better to add a brief paragraph in introduction about miRNAs, their role in cancer, chemoresistance and their regulation. Suggested article could be Doi, 10.1016/j.canlet.2021.03.025.
Author Response

(The authors gave the same response as above.)

Reviewer 4 Report
In this manuscript, the data show that knockdown SND1 increases the level of many siRNAs, especially several tumor suppressors in colon cancer cells. SND1-KD shows synergetic effects with drugs targeting MEK-ERK and Bcl-2 signaling pathways. They also developed a screening assay to identify the small compounds that inhibit RNA-binding function of SND1 and discovers a compound that inhibit the RNA binding capacity of SND1. This compound increases the expression of miR-1-3p and sensitizes cancer cells to Bcl-2 inhibitor. This manuscript is well organized and the finding is novel and important in this field.
There are some minor issues such as grammatical errors. Page 9, line 322. the sentence needs to be corrected.
Author Response

(The authors gave the same response as above.)

Reviewer 5 Report
The work by Lehmusvaara et al is original. The authors assessed the mechanism for SDN1 mediated stress tolerance in cancer cells by assessing 372 miRNAs which are thought to be associated with cancer progression. Likewise, the authors screened the novel chemical compounds which inhibits the SDN1 mediated miRNA destabilization, thereby they showed that RBP-miRNA interaction may be applied for novel cancer therapeutics. This manuscript is interesting and intriguing. However, there are some imprecisions in some parts. Followings are my specific comments.
Major
- MiRNA qPCR panel, RNeasy mini kit is not for suited for purification of small RNA, including miRNAs. The author should use RNA isolation kit applied for miRNA, such as miRNeasy kit and MirVana kits.
- Figure 1A, 3B, provide the n-number of western blotting and add the statistical significance.
- Figure 1B, provide the n-number and describe “n.s.” between each group.
- Figure1D-G, I think dDSS is not typically known method. To make it easier to understand the finding, the authors should provide details about dDSS in the main text or methods.
- Figure 4B, it seems about ~10 molecules diminished the SND1-RNA tracer binding. Why the authors focused only suramin, NF023, and PPNDS? Likewise, which plots show these molecules?
- The authors have not shown the direct binding of SDN1-miR1-3p, 582-5p and 340-5p. Confirm the binding by biological methods, such as RNA-immunoprecipitation.
- Figure 5D, why miR-340-5p and 582-5p were not increased by 50 and 100μM of suramin treatment. Table 1 shows the suramin inhibits the SDN1 and its domains-RNA tracer binding less than 10μ
- How was the cleavage of caspases in navitoclax, suramin and navitoclax+suramin group?
- The authors claimed that suramin mediated SDN1 inhibition rescue the tumor suppressive miR-1-3p degradation by SDN1, thereby sensitized to navitoclax. However, this statement is overstated because the authors did not assess the growth suppressive effects mediated by navitoclax influenced in miR-1-3p overexpressed or knocked-down cells.
Minor
- Methods, some titles should begin on a new lane and format should be in italic (e.g. lane 90, 140).
- miRNA qPCR, the authors claimed mir-103 is commonly used reference miRNA as it is normally stably expressed. Please add the reference manuscripts.
- Add the statistical assessment section in the methods.
Author Response

(The authors gave the same response as above.)

Round 2
Reviewer 1 Report
Unfortunately, the authors have not addressed any of my major points about the paper.
- The authors agree that suramin is not a good molecule with which to begin optimization.They argue that its importance is in proof of concept that the RNA binding function of SND1 can be inhibited with a small molecule that can have cellular effects. But they do not actually show that suramin prevents RNA binding, other than in the original FP screen. This is why I brought up this point, and here as well, the authors agree that the suramin-SND1 interaction is not very specific. So it is not clear to me what concept is being proven here: a very high concentration of a very non-specific inhibitor has cellular effects that have not been shown to even involve RNA binding.
- The authors raise the interesting possibility that “suramin prevents the degradation of miR-1-3p by inhibiting the binding of SND1 to miR1-3p.”In order to demonstrate that suramin is an inhibitor of the binding, they must at least show that SND1 actually binds miR-1-3p, and that suramin actually prevents this binding. Instead of providing experiments that demonstrate this hypothesis, they argue that SND1 has multiple functions, and that miRNA levels may be regulated in multiple ways. Here, too, we are left without any substantial proofs as to how SND1 functions or a direct connection to miR-1-3p.
- The distinction that the authors make between being dependent on miR-1-3p and being synergistic with miR-1-3p is not clear to me at all.If the increased sensitivity is a result of synergy between navitoclax and miR-13p, then that seems to me to argue that there is some dependence of the sensitivity on miR-1-3p. This may be direct or indirect, but inhibiting miR-1-3p expression should have some effect. On the other hand, if it does not have any effect, then this would argue against a synergistic role for miR-1-3p.
In the final analysis, the work on the sensitization of cells to chemotherapy after the reduction of SND1 is very nice and convincing. The screen for small molecule inhibitors is very preliminary, however, and the evidence provided is not even enough, in my opinion, for a proof of concept argument that SND1 can be effectively targeted by a small molecule inhibitor that will selectively inhibit its RNA binding ability.
Author Response
Unfortunately, the authors have not addressed any of my major points about the paper.
- The authors agree that suramin is not a good molecule with which to begin optimization.They argue that its importance is in proof of concept that the RNA binding function of SND1 can be inhibited with a small molecule that can have cellular effects. But they do not actually show that suramin prevents RNA binding, other than in the original FP screen. This is why I brought up this point, and here as well, the authors agree that the suramin-SND1 interaction is not very specific. So it is not clear to me what concept is being proven here: a very high concentration of a very non-specific inhibitor has cellular effects that have not been shown to even involve RNA binding.
We thank the reviewer for the clarified criticism. We have now performed additional experiments, which show that SND1 truly binds miR-1-3p and that suramin inhibits this binding (Supplementary figure 6 a and b).
- The authors raise the interesting possibility that “suramin prevents the degradation of miR-1-3p by inhibiting the binding of SND1 to miR1-3p.”In order to demonstrate that suramin is an inhibitor of the binding, they must at least show that SND1 actually binds miR-1-3p, and that suramin actually prevents this binding. Instead of providing experiments that demonstrate this hypothesis, they argue that SND1 has multiple functions, and that miRNA levels may be regulated in multiple ways. Here, too, we are left without any substantial proofs as to how SND1 functions or a direct connection to miR-1-3p.
This is also valuable comment. We indeed did these experiments to show that SND1 binds miR1-3-p and suramin (but not navitoclax) inhibits this treatment (supplementary figure 6).
- The distinction that the authors make between being dependent on miR-1-3p and being synergistic with miR-1-3p is not clear to me at all.If the increased sensitivity is a result of synergy between navitoclax and miR-13p, then that seems to me to argue that there is some dependence of the sensitivity on miR-1-3p. This may be direct or indirect, but inhibiting miR-1-3p expression should have some effect. On the other hand, if it does not have any effect, then this would argue against a synergistic role for miR-1-3p.
We thank the reviewer for valuable arguments. Argument about synergistic effect is based on the observation that silencing of SND1 or suramin treatment itself is not harmful for cancer cells. We hypothesize, based on previous publications, that SND1 enhances stress tolerance in cell. Thus, when navitoclax treatment increases chemical stress in cells, lack of SND1-miR-1-3p interaction leads to higher cell death.
In the final analysis, the work on the sensitization of cells to chemotherapy after the reduction of SND1 is very nice and convincing. The screen for small molecule inhibitors is very preliminary, however, and the evidence provided is not even enough, in my opinion, for a proof of concept argument that SND1 can be effectively targeted by a small molecule inhibitor that will selectively inhibit its RNA binding ability.
We agree that additional experiment to show the binding of SND1 to miR1-3-p and the inhibitory effect of suramin into this interaction was needed. We hope that this additional evidence gives more confidence to the proof-of-concept argument that SND1 RNA binding site is a potential target for small molecular inhibition.
Reviewer 2 Report
The authors seem to have been well addressed in the comments presented last time.
Author Response
The authors seem to have been well addressed in the comments presented last time.
We thank the reviewer for his/her time and effort to improve the manuscript.
Reviewer 5 Report
The revised manuscript is very improved. There are few comments to be addressed prior to publication.
The authors claimed the assessment of SND1 and miRNA binding is challenging. However, I don’t think so. For instance, the author assessed SND1-RNA binding by EMSA and FP assay in Figure 3. The assessment of SND1 and specific miR binding will be possible by using each synthetic Alexa labelled miR. Likewise, the inhibitory assay of miR-1-3p and SDN1 binding should be tested in navitoclax to exclude the secondary mechanism of miR-1-3p increasement.
Author Response
The revised manuscript is very improved. There are few comments to be addressed prior to publication.
The authors claimed the assessment of SND1 and miRNA binding is challenging. However, I don’t think so. For instance, the author assessed SND1-RNA binding by EMSA and FP assay in Figure 3. The assessment of SND1 and specific miR binding will be possible by using each synthetic Alexa labelled miR. Likewise, the inhibitory assay of miR-1-3p and SDN1 binding should be tested in navitoclax to exclude the secondary mechanism of miR-1-3p increasement.
We thank the reviewer for valuable comments and ideas. As suggested, we performed an FP assay to show that SND1 binds miR-1-3p, and suramin, but not navitoclax, inhibits the binding (supplementary figure 6).
Round 3
Reviewer 1 Report
The FP experiment that the authors have added (Supplemental Fig. 6) shows that suramin inhibition of SND1 binding to miR-1-3p has an IC50 of 1.4uM. Yet suramin has no significant affect on survival until the authors get to concentrations of 50 or 100uM. This seems to argue strongly that the inhibition of SND1 binding to miR-1-3p is not what is causing the sensitivity to navitoclax. Therefore, there is still no evidence, or even proof-of-concept that inhibition of RNA binding by SND1 is what is causing the increased sensitivity. Instead, the extremely high concentrations of suramin required for an effect would seem to be the most likely explanation.
Reviewer 5 Report
The authors adequately, if not completely, addressed my concerns. The manuscript is suited for publication in Cancers.
This manuscript is a resubmission of an earlier submission. The following is a list of the peer review reports and author responses from that submission.